# Study of Gliding Arc Plasma Treatment for Bamboo-Culm Surface Modification

**Bin Li †, Jinxing Li †, Xiaojian Zhou *, Jun Zhang, Taohong Li**  **and Guanben Du ***

Yunnan Provincial key Laboratory of wood Adhesive and Glued Products, Southwest Forestry University, Kunming 650224, China; touchbinbin@swfu.edu.cn (B.L.); jinxingli@swfu.edu.cn (J.L.); zj8101274@163.com (J.Z.); lith.cool@163.com (T.L.)

* Correspondence: xiaojianzhou@hotmail.com (X.Z.); gongben9@hotmail.com (G.D.)

† These authors contributed equally to this work.

**Abstract:** Plasma treatment was conducted to modify the outer- and inner-layer surfaces of bamboo in a multi-factor experiment, where the surface contact angles and surface energy were measured, followed by investigation on the surface microstructure and functional groups using a scanning electron microscope (SEM) and X-ray photoelectron spectroscopy (XPS), respectively. The result showed that when the power of the gliding arc plasma treatment was 1000 W while the bamboo surface was 3 cm away from the nozzle of the plasma thrower in the plasma flame, the contact angles of the outer- and inner-layer surfaces decreased, whereas the surface energy increased as a function of the treatment time. The 40 s treatment on the outer-layer surface caused the contact angle to reach 40°, and the surface energy accomplished a value of 45 J. Likewise, when the inner-layer surface was exposed for 30 s treatment, its contact angle attained a value of 15°, while the surface energy elevated to 60 J. Surface assessment with scanning electron microscopy (SEM) demonstrated etched microstructures of outer- and inner-layer surfaces of the bamboo culm after the treatment with gliding arc plasma. Moreover, the soaking test performed on the surfaces signified that 2D resin could have adhered more easily to outer- and inner-layer surfaces, which was considered a result of the greater uniformity and smoothness acquired after the treatment. X-ray photoelectron spectroscopic (XPS) analysis revealed that hydrophilic groups (O-CO-N, $-NO^{2-}$, $-NO^{3-}$, C-O-C, C-O-H and O-CO-OH, C-O-C = O) emerged on outer- and inner-layer surfaces of bamboo culms after being treated by gliding arc plasma, which enhanced the interaction of bamboo culms with applied protective coating resins.

**Keywords:** moso bamboo (*Phyllostachysheterocycle cv. Pubescens*); gliding arc plasma; surface treatment; activated surface; wettability

## 1. Introduction

Moso bamboo (*Phyllostachysheterocycle cv. Pubescens*) is widely used in architecture and furniture industries due to its fast-growing and good structural properties. However, the nutrient-rich parenchyma cells of bamboo, which contain starch and carbohydrates, make it highly susceptible to damages caused by molds and bamboo beetles, resulting in mildew, decay, and cracking. All these outcomes will reduce the service life of bamboo and limit its applications [1–3]. In addition, the existing green bamboo and yellow bamboo have a resulting poor gluability and paintability. There are some reports about the modification of bamboo using different methods [4–7], and a wide range of studies focusing on bamboo protection and modification have been undertaken from perspectives of their biodegradation mechanisms, preservatives development [8,9], and anti-decay methods [10–12].

The last few years have witnessed a considerable expansion of products made of bamboo culms. The absence of horizontal tissues, together with the presence of a wax layer on the hard surface, made

it very difficult for regular preservatives and mildew preventive treatments to penetrate smoothly into the bamboo. Further, other problems like the poor adherence of preservatives to bamboo culms and the susceptibility to cracking have not been addressed properly [13], rendering bamboo materials highly vulnerable to mildew and worm problems. Many antiseptic methods and equipment have been developed to address this issue. Among them, one is based on soaking preservatives by exerting pressure on one end of the bamboo [14]. This helped to effectively improve the decay resistance ability. However, these methods often involve utilizing complicated equipment, thus cannot be employed on a large-scale.

Plasma can be used to activate surfaces of materials by introducing new functional groups to the material surface, thereby enhancing materials' properties [15–19]. Exposing bamboo surface to plasma treatment results in enhanced surface wettability and improved the accessibility of coating materials [20,21]. However, most of undergoing research on the application of plasma for surface treatment of bamboo focuses on the mechanism of action and temporal effect of plasma. At present, plasma equipment, involving either radio frequency discharge (RF) [22] or dielectric barrier discharge (DBD) [23], are widely engaged in wood and bamboo treatments. Regrettably, the use of this equipment is restricted in terms of the sample sizes to be treated as they can only process small-sized materials [24,25].

Although gliding arc plasma treatment methods have been used in different fields [26–28], there is no research on bamboo treatment with gliding arc treatment. This shows the importance and challenge of using such existing technology to process large-sized materials like bamboo culms.

The 2D resin (Dimethylol dihydroxy ethylene urea, DMDHEU) is a typical resin of the nitrogen hydroxymethyl compound with low molecular weight, which has excellent properties in low formaldehyde emission, anti-wrinkle and anti-shrink mechanical properties, and stability, and is used to replace formaldehyde resin in the cotton fiber textile industry and wood modification [29–33]. Some researchers have carried out wood modification studies using 2D resin; the self-polymerization and cross-linking reaction of the 2D resin and wood compounds occurred within the cell wall, resulting in a permanent bulking of the cell wall and leading to a reduction in the swelling and shrinkage properties, thus with the dimensional stability considerably increased. In addition, good preservative properties and high resistance against white, brown, and soft rot fungi are obtained. The treatments also enhance the wood's acoustic, weathering, and aging properties, furnishing and gluing performances, as well as mechanical properties [30–33]. However, there are no reports on bamboo culms modification with the 2D resin.

Improving the retention of the 2D resin in bamboo was attempted in the current work after enhancement of the surface reactivity by applying its surfaces to low-temperature gliding arc plasma treatment, which would provide a theoretical basis and technical platform to solve the bamboo decay, mildew, instability and cracking, etc., for expanding the industrial use of bamboo.

## 2. Materials and Methods

### 2.1. Experimental Material

A 4-year-old fresh moso bamboo (*Phyllostachys heterocycle cv. Pubescens*) culm, with a height above 8 m, average diameter around 12 cm, minimum diameter no less than 8 cm, and wall thickness close to 10 mm, was selected from Longnan, Ganzhou City, Jiangxi Province, China. To prepare the samples for plasma treatment, bamboo culms with uniformly straight and smooth surfaces were initially cut 10 cm above ground level, then cut sequentially into 2 sections of 1.5 m bamboo tubes with a moisture content of 65%–70% and density of 700–720 kg/m$^3$. A gliding arc plasma equipment (model: CTD-2000 F; size: 250 × 200 × 360 mm; weight: 12 kg; handheld; Suman plasma, Nanjing, China) with an atmospheric low-temperature plasma torch was employed for surface treatment. It was fully controlled using software with one button through a microcontroller unit (MCU). The rated input power supply for the plasma torch was AC 220 V. Providing 1000 W output power, the plasma torch was capable of treating

specimens with a width between 60–80 mm. A lab-made 55%–65% active component dimethylol dihydroxy ethylene urea resin (DMDHEU, also known as 2D resin), with colorless to pale-yellow color and a slightly offensive odor, was used, which can be dissolved in water in any proportion. Analytical-grade diiodomethane and glycerol were utilized for contact angle measurements.

### 2.2. Plasma Treatment of Outer- and Inner-Layer of Bamboo Culms

In Figure 1, Bamboo culm specimens with the bamboo outer surface up were placed at different distances from the nozzle of the gliding arc plasma emitter (3, 5, and 7 cm), which was operated at a power of 1000 W. The moving speed of the thrower was set at 1 cm/s and the samples were treated repeatedly and evenly for 5, 10, 15, 20, 30, 40, and 60 s, respectively. The treatment of the inner layer or the inner wall surface was performed on bamboo culms cut into strips of 5 mm width and involved exposure to plasma radiation exactly as described for the outer layer.

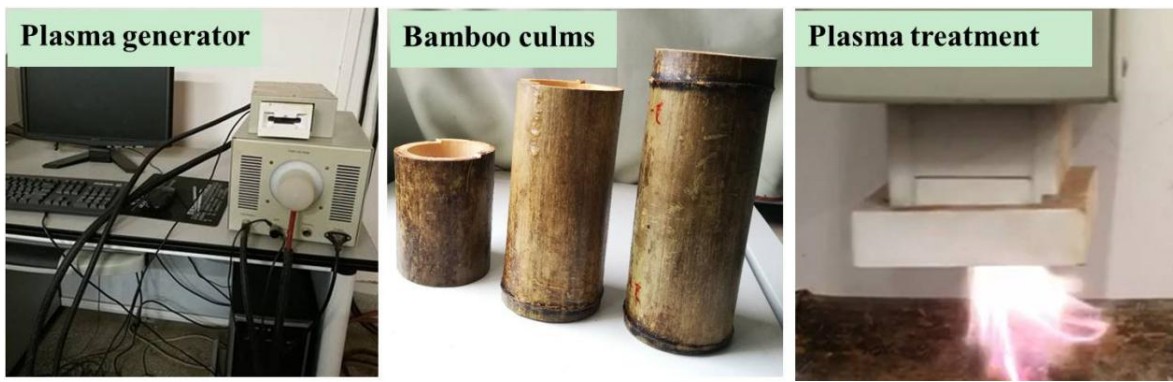

**Figure 1.** The plasma treatment of bamboo culms using a plasma generator.

### 2.3. Resin Soaking after Plasma Treatment

After treatment with cold plasma, the bamboo specimens, including culms and strips, were soaked with the 2D resin for 24 h. Subsequently, they were placed in a drying oven initially set at 60 °C, and the temperature was increased to 100 °C at a rate of 10 °C/min. Finally, the samples were taken out after the temperature subsided gradually to 40 °C.

### 2.4. Characterizations

#### 2.4.1. Contact Angle Measurements and Surface Energy Calculation

A static solvent contact angle meter (JC2000A, Shanghai Zhongcheng Co. Ltd., Shanghai, China) was utilized for measuring the contact angles of the outer- and inner-layer surfaces using diiodomethane and glycerol as test liquids in the laboratory condition of 25 °C and humidity of 40% RH. According to the sample preparation procedure, photos were taken 2 s after the liquids were dropped on treated bamboo surface, and the angle-measuring method was employed to measure the contact angles of the liquid on specimen surfaces. The measurements were undertaken on 6 points on the outer- as well as inner-layer surfaces of bamboo culms for each solvent, and average values were recorded. After that, the Owens two-liquid method was applied to calculate the surface energy values of the specimens [34].

#### 2.4.2. Surface Microstructure Observation

Both the outer- and inner-layer surfaces of treated bamboo specimens were fixed on the sample holder using conductive adhesive with the outer- or inner-layer to be tested facing upwards. Then, the specimens were placed in the compartment of sputter coating equipment, which was degassed using a vacuum pump for 5 min. The surfaces were sputter-coated with gold to avoid charging when the

vacuum level reached 3 Pa, and the surface morphology was examined using SEM (Quanta 200, ESEM, FEI, Hillsboro, OR, USA).

### 2.4.3. X-Ray Photoelectron Spectroscopy (XPS) Investigation

The treated outer- and inner-layer surfaces treated with gliding arc plasma were scanned with K-Alpha + X-ray Photoelectron Spectroscopy at a vacuum of almost $2 \times 10^{-7}$ mbar using monochromatic Al K$\alpha$ X-ray source with energy 1486.6 eV, 6 mA $\times$ 12 kV, and a spot size of 400 $\mu$m as the light beam. The measurement parameters were incorporated CAE (constant analyzer energy) as a scanning mode and a full-survey spectrum with a pass energy of 100 eV and step size of 1 eV or narrow-survey spectrum with a pass energy of 30 eV and step size of 0.1 eV. The binding energy measurements were calibrated according to C1s surface contamination standard (284.8 eV).

## 3. Results and Analysis

### 3.1. Effect of Plasma Treatment on the Surface Characteristics of Bamboo Culm

3.1.1. Effect of Gliding Arc Plasma Treatment of Bamboo Culm on Contact Angles and Surface Energy of Outer-Layer Surface

Figure 2a represents the trend of variation in the surface energy of the outer-layer surface of bamboo culm after exposure to the gliding arc plasma treatment for different time intervals, whereas the samples were mounted at various distances from the nozzle of the plasma emitter, 3, 5, and 7 cm. Initially, it is obvious that the 7 cm distance seems too far for the surface to be influenced by the treatment, and the change was statistically insignificant. The surface energy response in the case of the 5 cm distance was much higher, and maximum energy was achieved within 25 s and then started to decrease again with prolonging the treatment time to reach a minimal level. This may reveal either damage or carbonization of the surface. The 3 cm distance between the plasma emitter and the sample looks much more appropriate for the treatment, as the surface energy continued to increase with the exposure time up to 40 s. Even if the treatment was extended 20 s more, the extent of carbonization was very limited. Careful examination of the associated change in contact angle between bamboo culm and the resin indicates a strong correlation with the developed surface energies (Figure 2b). That is to say, if more active surface groups are born, they would contribute appreciably to enhance the wetting of the surface with the resin. Thus, the optimal contact angles between the bamboo culm and 2D resin can be obtained by verifying the samples exposed at 3 cm away from the nozzle of the plasma emitter while the treatment lasts for 40 s. Under such conditions, the contact angle between the outer-layer and the 2D resin declines remarkably from 100° to about 40°.

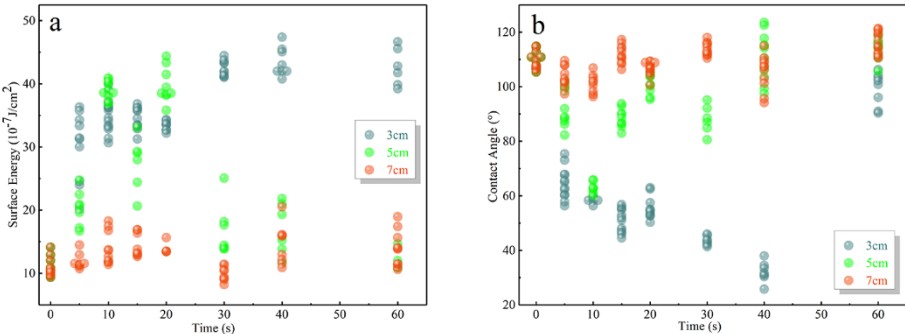

**Figure 2.** Variations in the surface energy of the outer-layer surface of bamboo culm exposed to treatment with gliding arc plasma (**a**) and the contact angle of the treated outer-layer with 2D resin (**b**) as a function of treatment time.

### 3.1.2. The Effect of Gliding Arc Plasma Modification on Inner-Layer Surface Properties

Figure 3 provides the trend of variations in surface energy and contact angle of the inner-layer after the gliding arc plasma treatment. It is clear that the surface energy of the inner-layer increased significantly regardless of the distance between the nozzle of the plasma emitter and the inner layer surface. However, the enhancement in surface energy is more sounding as far as the distance is shorter (Figure 3a). In parallel, the contact angle between the inner-layer surface and the 2D resin changes sharply in a reverse direction (Figure 3b). After 30 s, there might be a certain level of carbonization developed over time, which is the reason the contact angle started to re-increase.

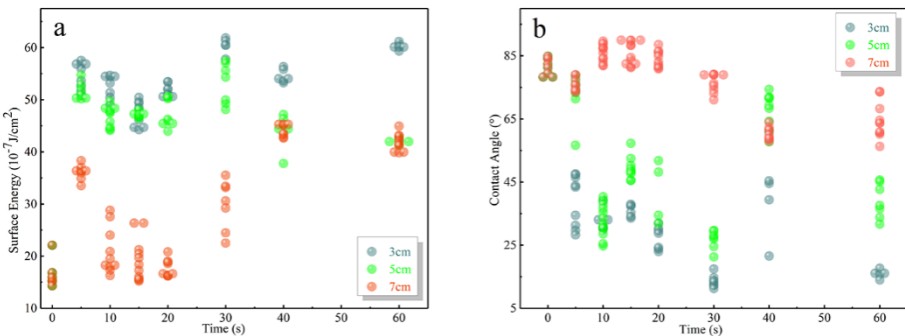

**Figure 3.** Variations in the surface energy of the inner-layer surface of bamboo culm exposed to gliding arc plasma treatment (**a**) and contact angle of the treated inner-layer with 2D resin (**b**) as a function of treatment time.

### 3.2. Effect of Plasma Treatment on Surface Microstructure of Bamboo Culm

### 3.2.1. Effect of Gliding Arc Plasma Treatment on the Microstructure of Outer-Layer Surface

As can be seen from the outer-layer surface microstructures shown in Figure 4, a significant difference can be recognized after the plasma treatment of 40 s, and the bamboo culm surface is around 3 cm away from the nozzle of the plasma emitter. Comparing Figure 4a with Figure 4d reveals etching developed on the outer-layer surface after gliding arc plasma treatment, which imposes the resin more liable for the soaking step. While comparing Figure 4b,e against Figure 4c,f dictates a uniform smoother surface, lacking any signs of cracking for the 2D resin soaking surface after the plasma treatment, which was not the case for the surface soaked with 2D resin without qualifying the surface to this step by prior plasma treatment. It is obvious for the latter that the 2D resin adhered more strongly to the outer-layer surface, which allows for an effective protective role for the resin on the bamboo culms.

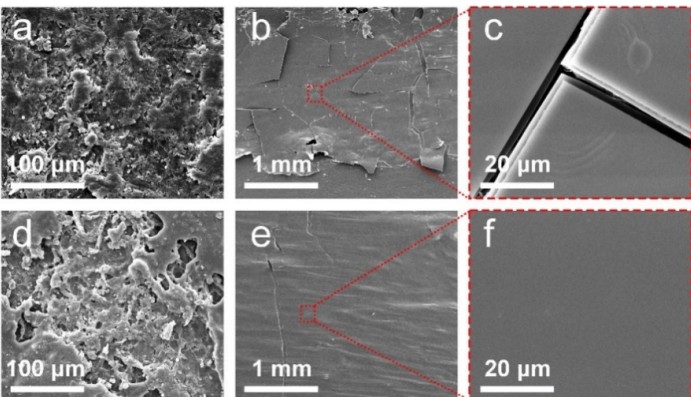

**Figure 4.** Morphological features of the outer-layer surface: (**a**) before resin soaking, (**b,c**) different magnification images after soaking followed by drying, (**d**) after exposing to gliding arc plasma treatment, and (**e,f**) different magnification images of soaking and drying following the treatment.

### 3.2.2. Effect of Gliding Arc Plasma Treatment on the Microstructure of Inner-Layer Surface

Figure 5 shows the morphological features of the inner-layer surface at different magnifications with the plasma treatment of 30 s, and the culm inner surface is around 3 cm away from the nozzle of the plasma emitter. Figure 5a,b exhibit the morphological aspects of the untreated inner-layer surfaces, which is relatively smooth, and no grains could be detected. On the other hand, the treatment with gliding arc plasma rendered the surface a little bit rougher (Figure 5d,e). The comparison between Figures 5c and 5f indicates that the spreading and covering of the 2D resin over the surface exposed in prior to plasma arc treatment proceeded much more effectively, which sheds light on the responsibility of the treatment in generating active sites that play the role of the receptor to the 2D resin and enhances its adhesion to the surface.

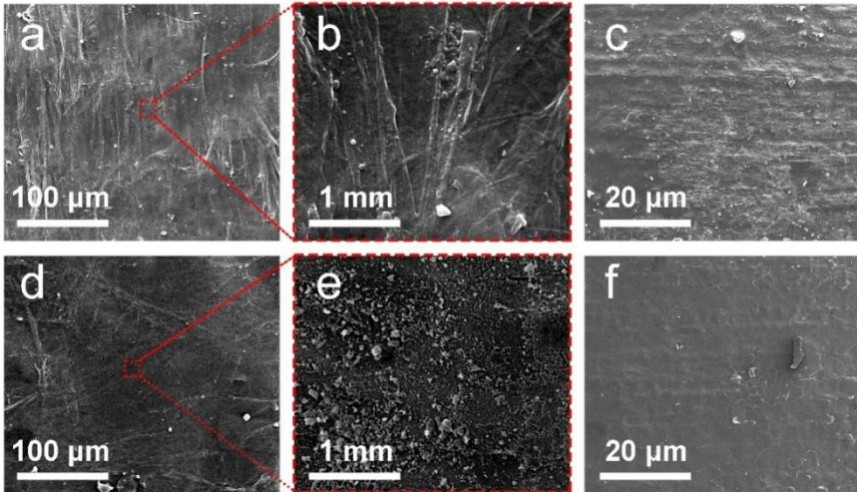

**Figure 5.** Morphological features of the inner-layer surface at different magnifications before (**a**,**b**) and after soaking followed by drying (**c**), different magnification images of the surface after exposing to gliding arc plasma treatment (**d**,**e**), and subsequent soaking and drying (**f**).

### 3.3. XPS Study of the Surface Functional Groups of Bamboo Culm Following Gliding Arc Plasma Treatment

#### 3.3.1. The Outer-Layer Surface

To verify changes on the outer-layer surface and evolution of polar groups after plasma treatment, XPS spectroscopic analysis was undertaken. Figure 6a–c shows the scans of some elements on the outer-layer surface before treatment with gliding arc plasma, which translates into a few active groups anchored on the surface. After exposure to plasma treatment, new active functional groups emerged on the outer-layer surface, as indicated mostly by changes in C1s environments (Figure 6d). This reveals the transformation of N-C = O into O-CO-N after the gliding arc plasma treatment. Similarly, some associated changes took place in the environments of the other elements, particularly O1s (Figure 6e,f), signifying the appearance of newborn hydrophilic groups. The emergence of these hydrophilic groups justifies the ability of 2D resin to be soaked and adhered more efficiently to the outer-layer surface of the bamboo culm.

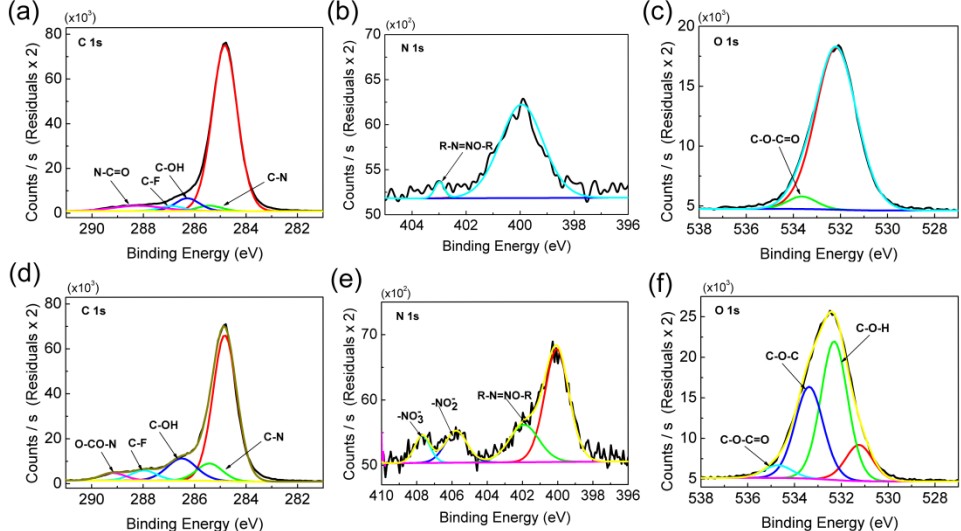

**Figure 6.** Narrow scan XPS spectra of the outer-layer surface for C1s (**a**), N1s (**b**), and O1s (**c**) elements before and after treatment (**d**–**f**), respectively.

### 3.3.2. The Inner-Layer Surface

Furthermore, Figure 7 exhibits the comparable XPS spectra recorded for the same groups on the inner-layer surface. It is clear that the extent of variation in the surface environment, especially the C1s, is much higher as compared to the outer surface. It additionally demonstrates that an additional hydrophilic O-CO-OH group appeared on the inner-layer surface after treatment. However, no significant difference was recognized in the case of N1s, whereas the corresponding scans of O1s show the emergence of an additional hydrophilic group C-O-C = O, which ensures enhanced accountability for resin soaking and penetration. The newly emerged groups on both inner as well as outer-layer surfaces of the bamboo culm are summarized in Table 1.

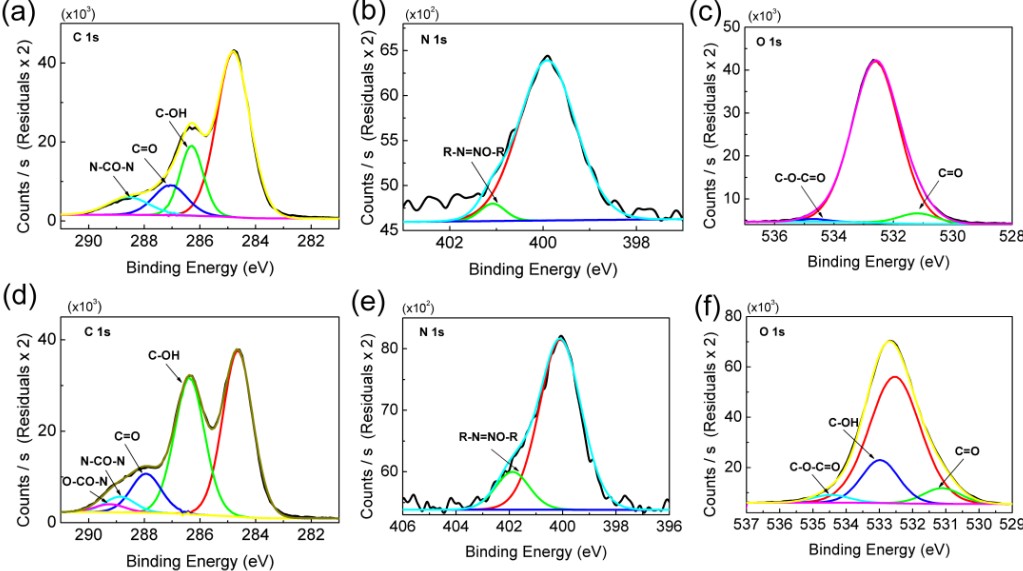

**Figure 7.** Narrow scan XPS spectra of the inner-layer surface for C1s (**a**), N1s (**b**), and O1s (**c**) elements before and after treatment (**d**–**f**), respectively.

**Table 1.** A list of the new-born hydrophilic groups on bamboo culm surfaces following gliding arc plasma treatment.

| Layers of Bamboo Culm | Evoluted Groups |
| --- | --- |
| Outer-layer | O-CO-N, $-NO^{2-}$, $-NO^{3-}$, C-O-C, C-O-H |
| Inner-layer | O-CO-OH, C-O-C = O |

Similar research results have been demonstrated in which new functional groups like $CF_3$, CHF, CF, C-O-H, C = O, COOH, $CO_2$, O-C-O, O = C-O, C = O, and C = N were found on the matrix surface after the plasma treatment [35–38]. This could be attributed to the formation of oxidized groups rich in hydroxyl, carbonyl, carboxyl groups, and phenoxy radicals. In this study, the evolution of functional groups can be explained by the fact that gliding arc plasma is a high energy ion, which can decompose the waxy material on the surface of bamboo so that the ester compounds in the waxy layer can be modified into new functional groups.

## 4. Conclusions

The outer- and inner-layer surfaces of bamboo-culm were activated by exposure to gliding arc plasma under optimized conditions. Maximized enhancement of the surface activity was reached when the bamboo culm surface was no more than 3 cm away from the nozzle of the plasma emitter for about 40 s. These conditions were quite sufficient, as revealed by the measurements of the contact angle and surface energy, 40° and 45 J, respectively. However, the inner-layer surface required shorter time (30 s) to attain the optimized state and became more energetic and wettable as demonstrated by the contact angle and surface energy, 15° and 60 J, respectively. This was achieved via etching and a morphological change of the surface, which enabled the surface to respond more effectively to soaking by 2D resin. The stronger adhesion achieved on the surface between the bamboo culm and 2D resin was a direct result of the emergence of new hydrophilic groups on the surface following the treatment, which caused more wetting and better resin spreading, leading to more effective protective covering of the surface.

**Author Contributions:** B.L. and J.L. contributed the experiment processing and characterization of the samples. J.Z. and T.L. contributed the XPS testing and analysis. X.Z. and G.D. contributed the design of the experiment and the analysis of the results as well as organize the whole manuscript.

**Funding:** This work was supported by the National Key Research and Development Program of China (Research of protection technology for bamboo culms, Project No. 2017YFD0600803), the National Natural Science Foundation of China (NSFC 31760187) and Yunnan Provincial Natural Science Foundation (2017FB060) as well as the Yunnan provincial youth and middle-age reserve talents of academic and technical leaders (2019HB026).

**Conflicts of Interest:** The authors declare no conflict of interest.

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
