# Peer review of "Study of Gliding Arc Plasma Treatment for Bamboo-Culm Surface Modification"

_forests, doi:10.3390/f10121086_

Round 1

Reviewer 1 Report

line 43 and subsequent: anti-corrosion treatments: please clarify. The word corrosion is normally used in combination with metals (oxidation)

fig 2+3: is the contactangle and surface energy without treatment included as well. The red line belonging to 7 cm seems to include 0 time. This is however unclear and should be described in the text.

fig 2+3: lines do not seem to fit the data accurately. Lines should either be removed or data about accuracy of fit (r2) should be provided.

section 3.2 and 3.3: results after exposing to gliding are plasma treatments are shown.However no information about treatment conditions (time / distance) are given. Since fig 2/3 show a clear impact of treatent conditions these should be mentioned here as well.

Author Response

line 43 and subsequent: anti-corrosion treatments: please clarify. The word corrosion is normally used in combination with metals (oxidation)

Response 1: Thank you for your suggestion, the sentences have been improved, especially the “corrosion” has been changed and improved.

fig 2+3: is the contactangle and surface energy without treatment included as well. The red line belonging to 7 cm seems to include 0 time. This is however unclear and should be described in the text.

Response 2: Yes, it is. I am very sorry for making you confused. Actually, “0 time” which mean the neat sample without any plasma treatment, which directly measure the contact angle and surface energy on the original bamboo culms, the related text was added in the text.

fig 2+3: lines do not seem to fit the data accurately. Lines should either be removed or data about accuracy of fit (r2) should be provided.

Response 3: Thank you for your advice, the lines are the trend lines, which mean the trend of the contact angle and surface energy on function of the treat times. Now, the lines are removed.

section 3.2 and 3.3: results after exposing to gliding are plasma treatments are shown. However no information about treatment conditions (time / distance) are given. Since fig 2/3 show a clear impact of treatment conditions these should be mentioned here as well.

Response 4: sorry for the missing information, the treatment conditions including time and distance are given in the red text at beginning of the paragraph for section 3.2 and 3.3.

Reviewer 2 Report

Plasma treatment of wooden surfaces is a very interesting topic both from a scientific and industrial practice point of view. The subject of this work was investigation of using of glinding arc plasma for bamboo-culm surface modification.

Paper requires some adjustments. Authors should answer the following questions and make changes and additions in the text: 

Abstract

Some information connected with the obtained results (conclusion) should be added. 

Keywords

There should be a wettability or contact angle keyword given. 

Introduction

Authors should give more information (papers) connected with poor gluability or paintability of bamboo wood.

Is plasma treatment necessary for such wood species?

Why Authors select such wood species for the measurement (own experiences, question from practice, literature data)? 

Materials and Methods

Better will be the chapter title ‘Materials and Methods” than Material and Method (Authors used more than 1 method).

Authors did not give information about basic properties of the bamboo wood (e.g. density, moisture content, porosity, etc.). These parameters influence in the significant manner on the wettability.

What about parameters (temperature, relative humidity) in the laboratory? What about conditioning time between testing of samples (plasma treatment and other investigations)?

Why Authors chose the plasma treatment time 5, 10, 15, 20, 30, 40 and 60s? Why they did not used 50 s?

Why diiodomethane and glycerol liquids were used for tests? Why Authors did not used distilled water (commonly used solvent for production proecological preservative agents, adhesives and lacquers)?  

Results and Analysis

The analysis of the obtained results in a correct way was carried out.  

Conclusions

The conclusion “…The stronger adhesion achieved on the surface between the bamboo culm and 2D resin was a direct result of the emergence of new hydrophilic groups on the surface following the treatment, which caused more wetting and better spreading, leading to more effective protective covering of the surface” may be given after deeper analysis - taking into account the anatomical structure (macro- and microscopic properties) of bamboo wood and adherence test. 

Paper may be published after minor additions and changes.

Author Response

Abstract

1. Some information connected with the obtained results (conclusion) should be added.

Response 1:Thank you for your suggestion, this part has been improved and connected with obtained results.  

Keywords

2. There should be a wettability or contact angle keyword given. 

Response 2:Thank you for your suggestion, the word “wettability” is added.

Introduction

3. Authors should give more information (papers) connected with poor gluability or paintability of bamboo wood.

Response 3: Thank you for your suggestion, this part has been improved and some information connected with poor gluability or paintability of bamboo along with the added relevant references (marked in red). By the way, all citations and references are now updated, new reference [7] from Forests was added.

4. Is plasma treatment necessary for such wood species?

Response 4: Yes, it is,  in the references of [15] and [18] have mentioned these information.

5. Why Authors select such wood species for the measurement (own experiences, question from practice, literature data)?

Response 5: Actually, according to our previous research experiences and industrial trial, the plasma treatment could be widely used in the wood industry. Bamboo has many similar performances compared to wood, thus, wood was selected to mention in this section.

Materials and Methods

6. Better will be the chapter title ‘Materials and Methods” than Material and Method (Authors used more than 1 method).

Response 6: Thank you for your suggestion, the “Materials and Methods” instead of “Material and Method”.

7. Authors did not give information about basic properties of the bamboo wood (e.g. density, moisture content, porosity, etc.). These parameters influence in the significant manner on the wettability.

Response 7: Thank you for your suggestion, the missing  information is  now added in the text (mark in red).

8. What about parameters (temperature, relative humidity) in the laboratory? What about conditioning time between testing of samples (plasma treatment and other investigations)?

Response 8: The laboratory is set at normal condition without any additional air condition, the plasma treatment condition has mentioned in the text, other investigations such as contact angle measurement condition was added in the text (mark in red).

9. Why Authors chose the plasma treatment time 5, 10, 15, 20, 30, 40 and 60s? Why they did not used 50 s?

Response 9: Actually, the variation trend of the contact angle and surface energy is very distinct, and 50 s treatment has been done in our experiment, but not show in the manuscript. We also can conclude the results without this data.

10. Why diiodomethane and glycerol liquids were used for tests? Why Authors did not used distilled water (commonly used solvent for production proecological preservative agents, adhesives and lacquers)?

Response 10: These two chemical agents are commonly used to measure the contact angle and calculate the surface energy, anyhow, our target agent used to bamboo is not neither these two agents, nor water, but 2D resin, their selection is only for compare and calculation.  Distill water is not selected because the hydrophilicity of the bamboo surface will affect the test results.

Results and Analysis

11. The analysis of the obtained results in a correct way was carried out.

Response 11: Thank you, but we also have improved few parts, please check the red text in this section.

Conclusions

12. The conclusion “…The stronger adhesion achieved on the surface between the bamboo culm and 2D resin was a direct result of the emergence of new hydrophilic groups on the surface following the treatment, which caused more wetting and better spreading, leading to more effective protective covering of the surface” may be given after deeper analysis - taking into account the anatomical structure (macro- and microscopic properties) of bamboo wood and adherence test.

Response 12: Thank you very much, it has been improved now.

Reviewer 3 Report

A more detailed introduction of the background is missing about the use of plasma technology on lignocellulosic materials. However, there is probably limited knowledge about its effects on bamboo, but there are numerous papers about its use on wood material for example. Furthermore, there is no detailed discussion from the use of different resins for impregnation of lignocellulosic materials.

It is not clear what was the reason of using the resin treatment on bamboo. You mention indirectly, that you want to protect the bamboo. But it is not clear, against what you want to protect it? Against decay or water or any other effect? Please specify the goals of the study more detailed!

It is not clear, why the 2D resin was chosen for the tests. Does it have any importance in bamboo modification? What are the improvements expected regarding the impregnation with this exact resin? Why not other resins? This resin is used for modifying textiles and wood for instance, but the reason of using this resin in this research is not cleared.

It is not cleared, why is it important to investigate the inner and outer surface of the bamboo separated? What are the main differences of the two surfaces?

In methods part, there is a subchapter "Resin Impregnation after Plasma Treatment", but in the description you write "bamboo specimens, including culms and strips, were soaked in 2D resin for 24h." Please clarify the method! Was it impregnation or soaking? Was that a surface treatment only, or a full cross section impregnation?

You measured the contact angle after 2 seconds of dropping the liquid on the surface. Did not you continue the measurement for longer period? It would be important to see how stable are the surface properties of the treated material.

In subchapter "2.3.2 Surface Microstructure Observation" you mention "galvanizing" of the specimens. What is it exactly? Do you mean sputter coating of the samples?

Results show, that surface energy and contact angle are influenced by the plasma treatment of the inner layer even from 7 cm distance, in opposit to the outer layer. What is the reason/background for that? Please describe!

What is the background of different reactions of the inner and outer layer of the bamboo material? Table 1. summarizes the new-born hydrophilic functional groups, but there is no explanation why those differences are present. Please describe the reasons and support the explanation by references!

Author Response

1. A more detailed introduction of the background is missing about the use of plasma technology on lignocellulosic materials. However, there is probably limited knowledge about its effects on bamboo, but there are numerous papers about its use on wood material for example. Furthermore, there is no detailed discussion from the use of different resins for impregnation of lignocellulosic materials.

Response 1: It has been improved the introduction section, please check the red text in the section. In this study, actually, only the 2D resin was used to impregnate the bamboo culms, diiodomethane and glycerol liquids are used to calculate the surface energy as reference.

2. It is not clear what was the reason of using the resin treatment on bamboo. You mention indirectly, that you want to protect the bamboo. But it is not clear, against what you want to protect it? Against decay or water or any other effect? Please specify the goals of the study more detailed!

Response 2:We have now improved and make it is clear, please check the red text in the section of introduction.

3. It is not clear, why the 2D resin was chosen for the tests. Does it have any importance in bamboo modification? What are the improvements expected regarding the impregnation with this exact resin? Why not other resins? This resin is used for modifying textiles and wood for instance, but the reason of using this resin in this research is not cleared.

Response 3: The utilization of 2D resin has been demonstrated in the introduction part (mark in red text).  The advantage of 2D resin and related references are added.

4. It is not cleared, why is it important to investigate the inner and outer surface of the bamboo separated? What are the main differences of the two surfaces?

Response 4: The inner surface is called bamboo inner skin, the outer surface is called bamboo green, which contain a wax layer will significantly affect the bamboo’s performances. Thus,  It is worthy to consider the both surface behaviors.

5. In methods part, there is a subchapter "Resin Impregnation after Plasma Treatment", but in the description you write "bamboo specimens, including culms and strips, were soaked in 2D resin for 24h." Please clarify the method! Was it impregnation or soaking? Was that a surface treatment only, or a full cross section impregnation?

Response 5: Thank you for your question, I notice that it is not clear and misunderstanding. I have now corrected the “impregnation” to “soaking” in all manuscript. The bamboo specimens are full soaking in the 2D resin.  

6. You measured the contact angle after 2 seconds of dropping the liquid on the surface. Did not you continue the measurement for longer period? It would be important to see how stable are the surface properties of the treated material.

Response 6: Actually, yes, we measured the contact angle after different times (1 s, 2 s, 3 s, 4 s,5 s, 6 s……), but only the contact angle value after 2 seconds was selected to show the in the manuscript. Because our main target is not investigate the variation trend of the contact angle with the time increase, but the contact angle value before and after modification by plasma treatment. Some other researches investigated the influence of the storage time after treatment by plasma on the materials.

7. In subchapter "2.3.2 Surface Microstructure Observation" you mention "galvanizing" of the specimens. What is it exactly? Do you mean sputter coating of the samples?

Response 7: The “galvanizing” mean “sputter coating”, I have corrected the related sentence in this section mark in red text, please check the manuscript.

8. Results show, that surface energy and contact angle are influenced by the plasma treatment of the inner layer even from 7 cm distance, in oppositto the outer layer. What is the reason/background for that?

Response 8:Thank you for your question, actually, the outer layer and inner layer have the similar trend, both layers obtained the good results of contact angle and surface energy within the 3 cm treatment distance.  

Please describe!

9. What is the background of different reactions of the inner and outer layer of the bamboo material? Table 1. summarizes the new-born hydrophilic functional groups, but there is no explanation why those differences are present. Please describe the reasons and support the explanation by references!

Response 9: Thank you for your suggestion, this section has been improved and given the reasonable explanation in the red text, and added new references.

Round 2

Reviewer 3 Report

Quality of the paper was adequately improved since the first round of review. Thus, it can be published in the Journal.